# Ankyrin-G and Its Binding Partners in Neurons: Orchestrating the Molecular Structure of the Axon Initial Segment

**DOI:** 10.3390/biom15060901

**Published:** 2025-06-19

**Authors:** Xiaowei Zhu, Yanyan Yu, Zhuqian Jiang, Yoshinori Otani, Masashi Fujitani

**Affiliations:** Department of Anatomy and Neuroscience, Faculty of Medicine, Shimane University, 89-1 Enya-cho, Izumo-shi 693-8501, Shimane, Japan; m249419@med.shimane-u.ac.jp (X.Z.); m249427@med.shimane-u.ac.jp (Y.Y.); m249606@med.shimane-u.ac.jp (Z.J.); yotani@med.shimane-u.ac.jp (Y.O.)

**Keywords:** Ankyrin-G, axon initial segment, proteome, binding partners

## Abstract

The axon initial segment (AIS) is a specialized subcellular domain that plays an essential role in action potential initiation and the diffusion barrier. A key organizer of the AIS is Ankyrin-G, a scaffolding protein responsible for clustering voltage-gated ion channels, cell adhesion molecules (CAMs), and cytoskeletal components at this critical neuronal domain. Recent proteomic analyses have revealed a complex network of proteins in the AIS, emphasizing Ankyrin-G’s central role in its molecular architecture. This review discusses new findings in the study of AIS-associated proteins. It explains how Ankyrin-G and its binding partners (such as ion channels, CAMs, spectrins, actin, and microtubule-associated proteins including end-binding protein 3, tripartite motif-containing protein 46, and calmodulin-regulated spectrin-associated protein 2) organize their structure. Understanding the dynamic regulation and molecular interactions within the AIS offers insights into neuronal excitability and reveals potential therapeutic targets for axonal dysfunction–related diseases. Through these dynamic interactions, Ankyrin-G ensures the proper alignment and dense clustering of key channel complexes, thereby maintaining the AIS’s distinctive molecular and functional identity. By further unraveling the complexity of Ankyrin-G’s interactome, our understanding of AIS formation, maintenance, and plasticity will be considerably enhanced, contributing to the elucidation of the pathogenesis of neurological and neuropsychiatric disorders.

## 1. Introduction

Neurons are the fundamental structural and functional units of the nervous system, responsible for transmitting signals throughout the neural networks. Typically, neurons comprise three primary components: the soma, axon, and dendrites. Although the soma and dendrites primarily receive and process incoming signals, the axon is crucial for propagating action potentials to target cells, enabling communication between neurons and other cell types.

A highly specialized axonal region, the axon initial segment (AIS), is located near the base of the axon, approximately 20–60 μm from the soma [1,2,3,4,5] (Figure 1). The AIS is the primary site for action potential initiation, a critical process for neuronal signal transmission. The high concentration of voltage-gated ion channels in the AIS significantly lowers the threshold for action potential generation, making it essential for both neuronal excitability and effective signal transmission [1,2,3,4,5].

Another crucial function of the AIS is acting as a barrier [6]. Although neuronal polarity forms rapidly during brain development, neurons must maintain this polarity over extended periods. To achieve this, neurons build a physical barrier that prevents the mixing of axonal and somatodendritic molecules, thereby preserving structural polarity. Indeed, the AIS blocks the diffusion of axonal membrane molecules, as demonstrated by fluorescent labeling experiments conducted during polarization [7].

Ankyrin-G is an important protein at the AIS that acts like a scaffold. It plays a central role in organizing various membrane proteins on the cell surface and cytoskeletal components [8,9,10]. By anchoring these elements, Ankyrin-G controls both the functional and structural properties of the AIS [11,12]. It interacts with key molecules, including voltage-gated sodium and potassium channels, the cytoskeletal protein βIV spectrin (Figure 2), and cell adhesion molecules (CAMs). These interactions are vital for maintaining neuronal excitability and ensuring efficient signal transmission. Disruptions in these molecular pathways are associated with the pathophysiology of numerous neurological disorders [13,14,15].

Identifying the complete protein expression profile of the AIS has historically been challenging owing to the detergent-insoluble nature of many AIS proteins, a consequence of their tight association with the Ankyrin-G-dependent cytoskeleton [16]. In addition, as the AIS is structurally continuous with both the soma and axon, isolating an AIS-specific proteome has proven difficult. Recently, biotin ligase-based proximity labeling methods have enabled the successful capture of the AIS proteome [16,17]. This review aimed to assess the current AIS proteomic data and highlight recent advances in understanding AIS proteins, particularly those interacting with Ankyrin-G, a critical AIS organizer.

## 2. AIS Proteome

Previous attempts to identify AIS-specific proteins using techniques such as yeast two-hybrid systems with Ankyrin-G [18], neurofascin [19,20], or the neuronal cell adhesion molecule (NrCAM) [21] as bait proteins have had limited success, largely due to the specialized structure of the AIS and the insolubility of AIS-associated proteins [16]. However, recent advances combining enzyme-mediated proximity-dependent biotinylation with quantitative mass spectrometry have enabled a more comprehensive analysis of the AIS proteome [16,17]. These studies identified 71 high-confidence AIS proteins and several other lower-confidence candidates (Table 1 and Appendix A) [17].

**Table 1 biomolecules-15-00901-t001:** AIS proteome analysis and localization analysis results.

Ran King	Previously Reported AIS Protein Names (Shadow: High Confidence AIS Proteins, White: Lower Confidence AIS Proteins)	Gene Names	AIS Localization	References
1	Tripartite motif-containing protein 46	*Trim46*	++	[22]
2	Spectrin beta chain, non-erythrocytic 4 isoform sigma6	*Sptbn4*	++	[23]
3	Ankyrin-3 (Ankyrin-G)	*Ank3*	++	[12]
4	Sodium channel protein type 2 subunit alpha (voltage-gated sodium channel subunit alpha Nav1.2)	*Scn2a*	++	[12]
5	Neurofascin	*Nfasc*	+	[24]
6	WD repeat-containing protein 7 (TGF-beta resistance-associated protein TRAG)	*Wdr7*	+	[17]
7	Tenascin-R (TN-R)	*Tnr*	+	[25]
8	Scribble planar cell polarity protein	*Scrib*	+	[17]
9	WD repeat domain 47	*Wdr47*	+	[17]
16	Band 4.1 (erythrocyte membrane protein band 4.1)	*Epb41 Epb4.1*	para-AIS	[26]
18	Versican core protein (chondroitin sulfate proteoglycan core protein 2)	*Vcan Cspg2*	+	[27]
20	Lissencephaly-1 protein (LIS-1)	*Lis-1*	+	[28]
23	Voltage-dependent P/Q-type calcium channel subunit alpha-1A	*Cacna1a*	Only physiologically certified	[29]
29	Brevican core protein	*Bcan Behab*	+	[24]
34	Sodium channel subunit beta-2	*Scn2b*	Other isoforms expressed	[30,31]
38	Glutamate decarboxylase 2 (GAD-65)	*Gad2 Gad65*	+	[32]
86	Potassium voltage-gated channel subfamily B member 1 (delayed rectifier potassium channel 1) (DRK1) (voltage-gated potassium channel subunit Kv2.1)	*Kcnb1*	+	[33]
165	PH and SEC7 domain-containing protein 1 (exchange factor for ADP-ribosylation factor guanine nucleotide factor 6) (exchange factor for ARF6)	*Psd Efa6*	+	[34]
183	Sodium/potassium-transporting ATPase subunit alpha-1 (Na(+)/K(+) ATPase alpha-1 subunit) (sodium pump subunit alpha-1)	*Atp1a1*	+	[17]
200	Voltage-dependent N-type calcium channel subunit alpha-1B (brain calcium channel III) (BIII) (calcium channel, L type, alpha-1 polypeptide isoform 5) (voltage-gated calcium channel subunit alpha Cav2.2)	*Cacna1b*	Only physiologically +	[29]
207	Potassium voltage-gated channel subfamily KQT member 2 (KQT-like 2) (potassium channel subunit alpha KvLQT2) (voltage-gated potassium channel subunit Kv7.2)	*Kcnq2*	+	[35]
228	Nuclear distribution protein nude-like 1 (Protein Nudel)	*Ndel1*	+	[28]
246	Voltage-gated potassium channel subunit beta-2 (K (+) channel subunit beta-2) (Kv-beta-2)	*Kcnab2*	+	[36]
269	Gamma-aminobutyric acid receptor subunit beta-1 (GABA (A) receptor subunit beta-1)	*Gabrb1*	Other isoforms +	[37]
355	Casein kinase II subunit alpha (CK II alpha)	*Csnk2a1*	+	[38]
371	F-actin monooxygenase	*Mical3*	+	[16]
455	Rho GTPase-activating protein 21	*Arhgap21*	+	[16]
470	Neuronal cell adhesion molecule (NrCAM)	*Nrcam*	+	[24]
507	Glycogen synthase kinase-3 beta (GSK-3 beta) (Factor A) (FA) (serine/threonine-protein kinase GSK3B)	*Gsk3b*	n/a	[39]
534	Gamma-aminobutyric acid receptor subunit beta-3 (GABA (A) receptor subunit beta-3)	*Gabrb3 Gabrb-3*	Other isoforms +	[37]
543	Inositol 1,4,5-trisphosphate receptor type 1 (IP3 receptor isoform 1) (IP-3-R) (IP3R 1) (InsP3R1) (type 1 inositol 1,4,5-trisphosphate receptor) (type 1 InsP3 receptor)	*Itpr1 Insp3r*	+	[40]
585	Leucine-rich repeat-containing protein 7 (Densin-180) (Densin) (Protein LAP1)	*Lrrc7 Lap1*	+	[41]
663	Microtubule-actin cross-linking factor 1 (actin cross-linking family 7)	*Macf1*	+	[16]
676	Neuroligin-2	*Nlgn2*	+	[42]
699	Potassium voltage-gated channel subfamily A member 2 (RAK) (RBK2) (RCK5) (voltage-gated potassium channel subunit Kv1.2)	*Kcna2*	+	[43]
709	Gamma-aminobutyric acid receptor subunit gamma-2 (GABA(A) receptor subunit gamma-2)	*Gabrg2*	Other isoforms +	[37]
748	Gamma-aminobutyric acid receptor subunit alpha-3 (GABA(A) receptor subunit alpha-3)	*Gabra3*	+	[37]
765	Tubulin alpha-4A chain (alpha-tubulin 4) (tubulin alpha-4 chain)	*Tuba4a*	n/a	[44]
768	Casein kinase 2 alpha 2 (casein kinase 2, alpha prime polypeptide)	*Csnk2a2*	n/a	[38]
800	Spectrin alpha chain, non-erythrocytic 1 (alpha-II spectrin) (fodrin alpha chain)	*Sptan1*	+	[45]
827	Septin-7 (CDC10 protein homolog)	*Septin7*	+	[16]
862	Calcium/calmodulin-dependent protein kinase type II subunit alpha	*Camk2a*	n/a	[46]
874	Disks large homolog 2 (channel-associated protein of synapse-110) (Chapsyn-110) (postsynaptic density protein PSD-93)	*Dlg2 Dlgh2*	+	[47]
985	Synaptopodin	*Synpo*	+	[48,49]
1000	Myc box-dependent-interacting protein 1 (Amphiphysin II) (Amphiphysin-like protein) (bridging integrator 1)	*Amph2*	+	[50]
1050	ADAM metallopeptidase domain 22	*Adam22*	+	[51]
1058	Septin-11	*Septin11*	+	[16]
1063	Calcium/calmodulin-dependent protein kinase type II subunit beta	*Camk2b*	n/a	[46]
1068	Microtubule-associated protein 6 (MAP-6) (145-kDa STOP) (STOP145) (stable tubule-only polypeptide) (STOP)	*Map6 Mtap6*	+	[52]
1271	Microtubule-associated protein 1A (MAP-1A) [cleaved into MAP1A heavy chain and MAP1 light chain LC2]	*Map1a Mtap1a*	+	[53]
1281	Microtubule-associated protein RP/EB family member 1 (APC-binding protein EB1) (end-binding protein 1) (EB1)	*Mapre1*	+	[54]
1307	Microtubule-associated protein RP/EB family member 3 (EB1 protein family member 3) (EB3)	*Mapre3*	+	[54]
1324	Kinesin-1 heavy chain (conventional kinesin heavy chain) (ubiquitous kinesin heavy chain) (UKHC)	*Kif5b Khc*	+	[55]
1371	Contactin-1 (neural cell surface protein F3)	*Cntn1*	+	[56]
1396	Tubulin beta-3 chain (neuron-specific class III beta-tubulin)	*Tubb3*	+	[57]

++ indicates specific expression of the protein in the AIS or NoR; + indicates detectable expression in the axon by immunohistochemical analysis. Table 1 includes high (shadow) and low (white)-confidence AIS proteins that have been previously reported. In addition to newly identified proteins (Appendix A), other previously reported proteins, such as Caspr2 [58], TAG1 [58], LGI1 [59], Nav1.6 [43], Nav1.1 [43], Kv1.1 [43], Kv1.2 [43], Kv1.4 [47], Navb4 [30], and Navb1 [31], IQCJ-SCHIP-1 [60], TRAAK [61,62], and GABARAP [18] are not listed here due to the limitation of proteome analysis. Table 1 and Appendix A are presented with the permission of Dr. Rasband.

Despite detecting highly abundant AIS proteins such as WDR47 and WDR7, subsequent validation using the gene-editing tool CRISPR confirmed that their localization was not restricted to the AIS. These proteins were also found in the neuronal soma, axons, or dendrites [17]. Moreover, protein 4.1B exists in the para-AIS and may be excluded from the AIS in motor neurons [26]. Therefore, accurate mapping still requires careful validation of individual protein localization due to the structural complexity and compartmentalization of the AIS.

## 3. Ankyrin-G

Ankyrins are crucial scaffolding proteins essential for cellular structural and functional organization, particularly in neurons. The vertebrate ankyrin family comprises three main isoforms: Ankyrin-R, Ankyrin-B, and Ankyrin-G, encoded by the genes *ANK1*, *ANK2*, and *ANK3*, respectively. These isoforms share common molecular structures [8].

In neurons, Ankyrin-G exists predominantly in three isoforms—190-kDa, 270-kDa, and 480-kDa—which are highly expressed in the brain [8,9,10]. Structurally, Ankyrin-G comprises several distinct domains: the ankyrin repeat domain (ARD) [63], the spectrin-binding domain (SBD) [64], the neuron-specific variable long insert (NSVLI), and the regulatory domain (RD) [65] (Figure 3). The ARD, located at the N-terminus, contains 24 ankyrin repeats that are critical for binding a wide array of membrane proteins, including ion channels, transporters, and CAMs, hence its designation as the membrane-binding domain [8,9,10]. The SBD, comprising two ZU5 domains and a UPA domain (collectively termed the ZZU tandem) [64], facilitates interactions with spectrins. These interactions link the ankyrin-membrane protein complex to the actin cytoskeleton, thereby maintaining AIS structural integrity and organizing membrane domains [8,9,10]. The NSVLI, found in the 270-kDa and 480-kDa isoforms, contains a serine-rich region derived from exon 37, which promotes the specific localization of these isoforms to the AIS and nodes of Ranvier (NoR) [8,9,10]. The RD comprises a conserved death domain and an unstructured region that modulates interactions with other domains, potentially regulating ankyrin’s ability to bind target proteins through an autoinhibition mechanism [8,9,10].

**Figure 3 biomolecules-15-00901-f003:**
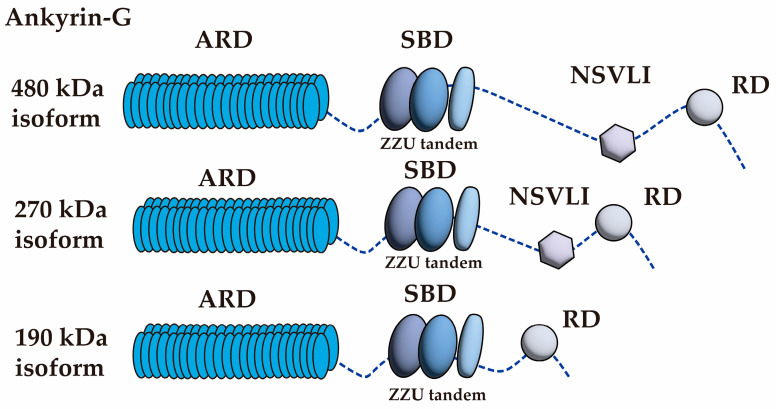
Protein structure of all isoforms of Ankyrin-G. Ankyrin-G consists of four distinct domains: ARD, the ankyrin repeat domain; SBD, the spectrin-binding domain; ZZU, two ZU5 domains followed by a UPA domain; NSVLI, the neuron-specific variable long insert; and RD, the regulatory domain.

The 190-kDa isoform of Ankyrin-G, which includes the ARD, SBD, and RD, is mostly found in the cytoplasm, postsynaptic sites, and dendritic spines, where it influences synaptic transmission and spine morphology [66]. This isoform is essential for synaptic plasticity and is a key component of the postsynaptic densities in rodents [66]. In contrast, the 270-kDa and 480-kDa isoforms, which include the NSVLI, are specifically localized to the AIS and NoR [67]. The 270-kDa isoform shows predominant expression in the frontal and cingulate cortices, while the 480-kDa isoform is more abundantly expressed in the cerebellum [68]. These larger isoforms are important for arranging proteins and synaptic structures at the AIS and NoR, playing integral roles in maintaining neuronal excitability [67].

Ankyrin-G expression in the brain is tightly regulated during development. The 480-kDa isoform is highly expressed in the AIS during early postnatal development but declines with age, whereas the 270-kDa isoform is consistently expressed throughout life [69]. The 190-kDa isoform increases from birth and reaches stable expression in adulthood [9]. Within the AIS, Ankyrin-G binds various proteins through multiple domains (Table 2). The ARD interacts with ion channels, transporters, and CAMs, while the SBD connects to spectrins, enabling the linkage of membrane proteins to the cytoskeleton. The NSVLI and RD provide additional binding capacity, thereby expanding Ankyrin-G’s range of molecular interactions [8,9,10].

In brief, Ankyrin-G serves as a multifunctional scaffold that links membrane proteins to the cytoskeleton through its distinct domains, with its isoform-specific expression and localization playing essential roles in AIS formation, maintenance, and neuronal excitability.

**Table 2 biomolecules-15-00901-t002:** Ankyrin-G binding partners at AIS.

AIS Localized Protein Names	Binding Domain of Ankyrin-G	Method	References
Nav1.2 and Nav1.6	N/A	They determined the AIS localization signal at the cytoplasmic II–III region of Nav1.2	[70]
	ARD	Pull-down assay	[71]
Kv7.2 and Kv7.3 (KCNQ2/3)		Reduced rate of fluorescence recovery after photobleaching (FRAP)	[72]
Neurofascin		Co-immunoprecipitation and in vitro ankyrin-binding assays	[73]
Neurofascin, NrCAM		Immunoblotting analysis and immunofluorescence staining	[74]
KIF5 (KIF5B)		Pull-down assay	[55]
βIV spectrin	SBD SBD (Zu5)	Co-immunoprecipitation	[23]
		Co-immunoprecipitation	[75]
		Surface plasmon resonance	[76]
Ndel1	NSVLI	Isothermal Titration Calorimetry (ITC) assays	[77]
GABARAP		Proximity ligation assay	[18]
IQCJ-SCHIP-1	ARD	Pull-down assay	[60]
TRAAK (K2P4.1)	N/A	Single-molecule pull-down assay Co-immunoprecipitation	[61] [62]
EB1/EB3	RD	Pull-down assay	[54]

Ankyrin-G forms four distinct domains, as described in Figure 3, and each domain interacts with distinct ion channels, transporters, CAMs, signaling molecules, and cytoskeletal elements. N/A: not available; ARD: the ankyrin repeat domain; SBD: the spectrin-binding domain; NSVLI: the neuron-specific variable long insert; RD: the regulatory domain.

## 4. Ion Channels/Transporters

Sodium channels are highly concentrated at the AIS; some studies suggest that their density is 5–50 times greater than in more distal axonal or dendritic regions [8,9,10,78]. This elevated concentration considerably reduces the membrane potential required to initiate an action potential [79]. Ankyrin-G is crucial for clustering these sodium channels at the AIS [8,14].

Sodium channels in mammals consist of a main pore-forming α-subunit and auxiliary β-subunits. The α-subunit comprises four domains (I–IV), each with six transmembrane segments. A critical feature is an intracellular loop connecting domains II and III, which includes a specific nine-amino-acid motif ([V/A]P[I/L]AXXE[S/D]D) required for Ankyrin-G binding [70,71,72]. Moreover, sodium channel localization at the AIS and NoR is modulated by casein kinase 2 (CK2), an enzyme highly enriched in these regions in both cortical and hippocampal neurons [38]. CK2 phosphorylates serine residues within the sodium channels, enhancing their affinity for Ankyrin-G. Inhibition of CK2 reduces sodium channel density at the AIS [38]. However, no evidence currently supports direct phosphorylation of Ankyrin-G by CK2 [80]. Furthermore, recent studies show that the localization of IQ motif-containing J-Schwannomin-Interacting Protein 1 (IQCJ-SCHIP-1) at the AIS depends on its phosphorylation by CK2 and subsequent interaction with Ankyrin-G, suggesting a novel AIS maintenance mechanism [81].

The main sodium channel subtypes at the AIS include voltage-gated sodium channel (Nav) 1.1, Nav1.2, and Nav1.6 [82,83]. These subtypes show distinct expression patterns depending on neuronal type and developmental stage. In early development, Nav1.2 predominates, while Nav1.6 becomes more prevalent as neurons mature [82]. In cortical neurons, Nav1.2 (with a higher activation threshold) localizes primarily to the proximal AIS, whereas Nav1.6 (with a lower activation threshold) is enriched in the distal AIS [84]. This spatial separation suggests subtype-specific roles in action potential initiation and propagation.

In addition to sodium channels, potassium channels also play a crucial role in shaping action potentials at the AIS. The voltage-gated potassium channel (Kv) 7.2 and Kv7.3 (KCNQ2/3) channels, which can form homomeric or heteromeric complexes, directly interact with Ankyrin-G at the AIS [72]. These channels contain a nine-amino-acid motif ([I/L] AX [E/V/L] GE[S/T] DX[T/F/W] E/D) responsible for Ankyrin-G binding [72]. Other potassium channels, such as Kv1.1 and Kv1.2, also accumulate at the AIS but do not directly bind to Ankyrin-G [43,47]. Instead, they are believed to localize in the distal AIS via interaction with the scaffolding protein postsynaptic density-93 (PSD-93) (Figure 2) [47]. Recent findings have also identified the TWIK-related arachidonic acid-activated potassium (TRAAK) channel (K2P4.1) as an AIS component, exhibiting periodic nanoscale co-distribution with Ankyrin-G in excitatory neurons [61,62].

The AIS also receives gamma-aminobutyric acid (GABA)-ergic innervation from specific interneurons [85,86]. Notably, chandelier cells form arrays of GABAergic synapses along the AIS in the cortex and hippocampus, while basket cells target the AIS of cerebellar Purkinje neurons [87,88]. A specific GABA_A receptor subtype containing the α2 subunit (α2-GABA_AR) is enriched at AIS axo-axonic synapses, anchored by postsynaptic scaffolding proteins gephyrin and collybistin and associated with Kv2.1 channels [89]. Furthermore, the adhesion molecule contactin-1 (Cntn1) is required for axo-axonic innervation [56]. A rare Ankyrin-G mutation (W1989R), found in a family with bipolar disorder, disrupts Ankyrin-G/GABA_A receptor-associated protein (GABARAP) interactions, leading to reduced GABAergic synapse numbers at both the AIS and somatodendritic domains of pyramidal neurons (Figure 2) [90]. This finding underscored Ankyrin-G’s crucial roles beyond the AIS.

Other transporters, including anion exchanger 1 (AE1), the ammonium transporter Rh type B [91], and the sodium/potassium-transporting ATPase subunit alpha-1, have also been reported to interact with Ankyrin-G [17,92]. However, their specific localization to the AIS has not been conclusively demonstrated [9,17,78].

In essence, the AIS harbors a complex array of ion channels and transporters whose precise localization and interactions with Ankyrin-G and associated regulatory proteins critically shape action potential initiation, neuronal excitability, and synaptic integration.

## 5. Spectrins

The spectrin cytoskeleton comprises heterotetramers of two α and two β subunits located just beneath the plasma membrane of the AIS [1,10,93]. Brain-specific assemblies are formed by βI, βII, βIII, and βIV spectrins, with the αII spectrin being the sole α isoform present in the brain [1,10,93]. In the AIS, the spectrin tetramer contains the βIV and αII subunits (Figure 2) [94]. βIV spectrin has two splice variants (Σ1 and Σ6), and the 280 kDa full-length isoform, βIVΣ1, containing an actin-binding domain, 17 triple-helical spectrin repeats, and a specific pleckstrin homology domain [10]. The longest isoform is proposed to bind the actin cytoskeleton through an actin-binding domain at the N-terminus, while the shorter truncated form, βIVΣ6, consists only of the C-terminal half of the protein, starting from the spectrin repeat. Both isoforms contain 14–15 spectrin repeats, enabling them to bind Ankyrin-G and localize to the AIS but only the Σ1 variant contains the canonical actin-binding domain [10]. Notably, βIVΣ1 is predominantly expressed during development, while βIVΣ6 becomes the major splice variant in mature neurons [69]. The mechanism by which the βIVΣ6 isoform binds actin remains unclear [93].

A pair of α/β spectrins binds side-by-side to form a heterodimer, forming a periodic cytoskeleton at the AIS and NoR, as revealed by the three-dimensional stochastic optical reconstruction microscopy (3D-STORM) imaging [45]. Conditional knockout studies reveal that αII spectrin is essential for maintaining the AIS structure, as its loss results in a fragmented AIS architecture [45]. Similarly, a deficiency of βIV spectrin in the brain leads to the disruption of the AIS structure in vivo [23].

In sum, βIV spectrin and αII spectrin form a periodic cytoskeletal lattice at the AIS that is essential for its structural integrity. Their dynamic regulation and interaction with Ankyrin-G support both the assembly and maintenance of the AIS architecture.

## 6. Actin Cytoskeleton and the Related Proteins

Actin filaments are essential structural components of the AIS submembrane cytoskeleton, forming a periodic ring-like lattice beneath the plasma membrane (Figure 2), known as the membrane-associated periodic skeleton (MPS) [44,95]. These circumferential actin rings are spaced approximately 190 nm apart and are believed to be crosslinked by βIV spectrin and αII spectrin tetramers, which align longitudinally along the AIS [44,95]. In this arrangement, the actin-binding N-termini of βIV spectrin colocalize with the actin rings, while their C-termini face inward and serve as binding sites for Ankyrin-G [44,95]. In addition to ring structures, actin filaments form patches and bundles within the AIS and axon, potentially preventing dendritic proteins from entering the AIS and acting as a vesicular filter [96]. Recent evidence indicates that the number of longitudinal actin filaments increases following plasticity induction and that actin polymerization mediated by formin family proteins is required for both AIS plasticity and the formation of longitudinal actin fibers [97].

The phosphorylated myosin light chain (pMLC), an activator of contractile myosin II, is highly enriched in assembling and maturing AIS, where it associates with actin rings (Figure 2) [98]. MLC phosphorylation and myosin II contractile activity are necessary for the AIS assembly and regulation of AIS component distribution along the axon [98]. Tpm3.1 exhibits a periodic distribution similar to submembrane actin rings (Figure 2), although it partially overlaps with them. Inhibition of Tpm3.1 leads to a decreased accumulation of AIS structural and functional proteins, disrupted sorting of somatodendritic and axonal proteins, and reduced neuronal firing frequency [99].

Ankyrin-G plays a central role in linking the submembrane actin–spectrin network to membrane proteins and possibly to deeper cytoskeletal components. Its N-terminal domain binds βIV spectrin near the C-terminus of spectrin tetramers, reinforcing the periodic arrangement of the spectrin–actin lattice (Figure 2) [44]. The 3D-STORM imaging shows that Ankyrin-G itself exhibits periodicity that mirrors that of βIV spectrin and actin, particularly in the N-terminal spectrin-binding region, whereas the C-terminal tail is more irregular and projects deeper into the AIS shaft [44]. This extension enables Ankyrin-G to interface with microtubules and other cytoplasmic components, acting as a bridge between the membrane cytoskeleton and the AIS interior [44].

Taken together, actin filaments and their associated proteins form a complex and adaptable scaffold that cooperates with spectrins and Ankyrin-G to regulate the AIS structure, plasticity, and selective trafficking of membrane components.

## 7. Microtubule-Associated Proteins

The AIS is characterized by tightly bundled fascicles of microtubules, which are essential for maintaining neuronal polarity and the separation of axonal and somatodendritic compartments [5,6]. The discovery of AIS by electron microscopy revealed it to be a microtubule-rich domain [100,101]. The formation and maintenance of these microtubule bundles are highly dependent on Ankyrin-G, as Ankyrin-G knockout mice exhibit a complete loss of AIS microtubule bundles and the dense membrane undercoat typically observed in this region [102].

Several key microtubule-associated proteins (MAPs) localize to the AIS and contribute to its unique cytoskeletal architecture (Table 1, Figure 2). End-binding proteins 1 and 3 (EB1/3) are critical plus-end tracking proteins that interact with Ankyrin-G via specific motifs located in Ankyrin-G’s C-terminal tail domain [54,103,104]. These interactions help anchor microtubules to the plasma membrane of the AIS, thus stabilizing the overall structure. Recent evidence highlights the pathological significance of EB3, showing that frontotemporal dementia causing the V337M tau mutation impairs AIS plasticity through the abnormal accumulation of EB3 in the submembrane region. This EB3 accumulation leads to excessive neuronal activity under chronic depolarization [105]. In addition, Ndel1, a dynein regulator, is enriched at the AIS and binds the NSVLI of Ankyrin-G, contributing to polarized cargo transport along microtubules [28].

Although there is no evidence that Ankyrin-G directly binds to the following proteins, calmodulin-regulated spectrin-associated protein 2 (CAMSAP2) anchors the minus ends of microtubules at the AIS, ensuring their stability [106]. Microtubule affinity-regulating kinase 2 also localizes to the AIS and the proximal axon enrichment zone (PAEZ), regulating tau association with microtubules and modulating their dynamics [107]. Microtubule cross-linking factor 1 (MTCL1) in cerebellar Purkinje cells mediates the formation of stable microtubule bundles, which are crucial for maintaining Ankyrin-G localization [108].

Another key protein is tripartite motif-containing protein 46 (TRIM46), which facilitates the formation of parallel microtubule bundles in the AIS [22,109,110]. This association promotes the fasciculation of microtubules into tightly aligned arrays. Notably, TRIM46 accumulates during neuronal development before Ankyrin-G clustering, particularly at the transition from stage 3 to stage 4, and extends into PAEZ, suggesting a role in early AIS assembly [111]. However, recent evidence suggests that TRIM46 is required for microtubule fasciculation in vivo but not AIS formation, as shown in knockout mice studies [109]. The acute deprivation of TRIM46 via shRNA [22,110] and the knockout model may yield differing results, warranting further investigation [109].

Together, these proteins, in concert with Ankyrin-G, organize and stabilize the microtubule cytoskeleton at the AIS. This specialized structure supports the functional compartmentalization of neurons and provides a scaffold for the trafficking of ion channels and other proteins critical for action potential initiation. The AIS, along with the PAEZ, acts as a major microtubule-organizing center, integrating signals from multiple MAPs to maintain neuronal polarity and excitability.

## 8. Conclusions

The AIS is a specialized neuronal domain crucial for action potential initiation and the maintenance of neuronal polarity. Ankyrin-G orchestrates the assembly of ion channels, adhesion molecules, and cytoskeletal components, defining AIS identity. Despite advances, key mechanisms regulating AIS remodeling and Ankyrin-G dynamics under physiological and pathological conditions remain unclear. Further research integrating advanced imaging, proteomics, and gene editing will be essential to unravel these processes and may reveal new therapeutic targets for neurological disorders involving AIS dysfunction.

## Figures and Tables

**Figure 1 biomolecules-15-00901-f001:**
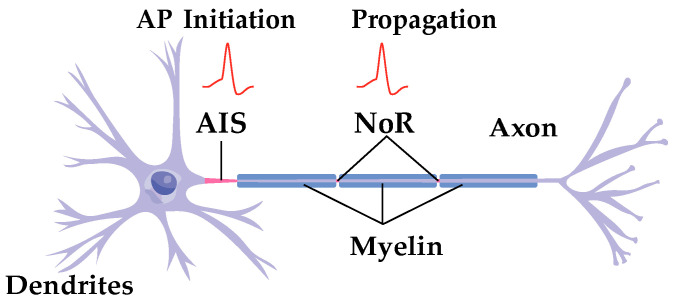
Axon initial segment (AIS) and node of Ranvier (NoR). The AIS is located at the proximal part of the axon and is responsible for initiating the action potential (AP). The NoR is a site of AP propagation.

**Figure 2 biomolecules-15-00901-f002:**
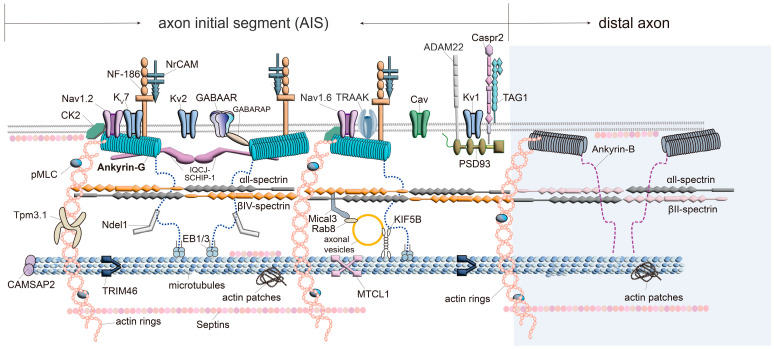
AIS molecular structure. Membrane proteins (ion channels and CAMs) are anchored by the N-terminal domain of Ankyrin-G (sky blue). The C-terminal of Ankyrin-G is inserted into spectrin tetramers. Within the AIS, αII spectrin (gray) and βIV spectrin (orange) form tetramers, whereas in regions distal to the AIS, αII spectrin (gray) pairs with βII spectrin (pink) to form tetramers. Actin rings (pink) are connected with the spectrin tetramers. In the distal part of the AIS, Kv1 channels, ADAM22, Tag1, and Caspr2 are present in certain neurons. Ankyrin-G binds to microtubules via EB1/EB3 proteins and Ndel1. Microtubule bundles are crosslinked by TRIM46 and MTCL1. CK2, MLC, and Tpm 3.1 bind to actin rings. Mical3 and septins are also present at the AIS and are involved in regulating the AIS structure. Abbreviations: AIS, axon initial segment; ADAM22, a disintegrin and metallopeptidase domain 22; CAMS, cell adhesion molecules; CAMSAP2, calmodulin-regulated spectrin-associated protein 2; Caspr2, contactin-associated protein-like 2; Cav, voltage-gated calcium channel; CK2, casein kinase 2; EB1/EB3, end-binding proteins 1 and 3; GABAAR, gamma-aminobutyric acid type A receptor; GABARAP, gamma-aminobutyric acid receptor associated protein; IQCJ-SCHIP-1, IQ motif-containing J-Schwannomin-Interacting Protein 1; KIF5B, Kinesin family member 5B; Kv, voltage-gated potassium channels; Mical3, molecule interacting with CasL 3; MLC, myosin light chain; MTCL1, microtubule cross-linking factor 1; Nav, voltage-gated sodium channels; Ndel1, nuclear distribution element-like 1; NF-186, neurofascin-186; NrCAM, neuronal cell adhesion molecule; PSD-93, postsynaptic density 93; Tag1, transient axonal glycoprotein 1; Tpm 3.1, tropomyosin 3.1; TRAAK, TWIK-related arachidonic acid-activated potassium; and TRIM46, tripartite motif-containing protein 46.

## Data Availability

Not applicable.

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
