# Peer review of "Ankyrin-G and Its Binding Partners in Neurons: Orchestrating the Molecular Structure of the Axon Initial Segment"

_biomolecules, 2025, doi:10.3390/biom15060901_

Round 1
Reviewer 1 Report
Comments and Suggestions for Authors
The article is a well-structured and informative review focusing on the axon initial segment (AIS) and its key organizer, Ankyrin-G. It comprehensively covers the molecular architecture of the AIS, its role in action potential initiation, neuronal polarity maintenance, and its implications in neurodegenerative diseases. The content is logically organized, supported by up-to-date research, and will be valuable for neuroscientists studying neuronal excitability mechanisms. The article thoroughly examines AIS structure, including ion channels, cytoskeletal proteins, and cell adhesion molecules. The author references modern techniques (e.g., proximity labeling, CRISPR, 3D-STORM) and recent publications. Schematics (Figures 1–3) and tables (Tables 1–2) enhance the understanding of complex information.
Some sections are overly detailed (e.g., exhaustive protein lists in Table 1), which may hinder readability. Brief summaries at the end of subsections would improve flow.
The manuscript is recommended for publication after minor revisions.
Author Response
Comment 1: Some sections are overly detailed (e.g., exhaustive protein lists in Table 1), which may hinder readability.
Response: We appreciate the reviewer’s valuable feedback regarding the level of detail in Table 1. In response, we have carefully revised and reorganized Table 1 to improve its clarity and readability.
Comment 2:Brief summaries at the end of subsections would improve flow.
Response:
We thank the reviewer for this insightful suggestion. In response, we have added concise summary sentences at the end of relevant subsections to improve the logical flow and aid reader comprehension. (Line 112, 160, 226, 250, 284)
Reviewer 2 Report
Comments and Suggestions for Authors
The review by Zhu et al. is a balanced presentation of current knowledge of the AIS proteome. I have only a few suggestions for improvement:
- When discussing the AIS proteome the authors state: “Although the first study had certain limitations, the results from the second study are considered more reliable as they encompassed a broader range of AIS-specific proteins.” I think this sentence should be removed. All studies have limitations and both of these proteomic studies have limitations. The range of the bioID is a limitation, while the longer range of HRP is also a limitation as it would tend to lead to higher numbers of false positives. I recommend simply removing this sentence as there does not need to be a comparison between the two. Both studies revealed new AIS proteins and were a first start at defining the AIS proteome.
- The reference for Kcnab2 in the list is incorrect. The referenced paper does not deal with Kvb2, but rather Kv2.1.
- I suggest including the TRAAK/TREK-1 two-pore potassium channel in Fig. 1, and include the reference from Escobedo et al. (2024), JCB.
Author Response
Comment 1: When discussing the AIS proteome the authors state: “Although the first study had certain limitations, the results from the second study are considered more reliable as they encompassed a broader range of AIS-specific proteins.” I think this sentence should be removed. All studies have limitations and both of these proteomic studies have limitations. The range of the bioID is a limitation, while the longer range of HRP is also a limitation as it would tend to lead to higher numbers of false positives. I recommend simply removing this sentence as there does not need to be a comparison between the two. Both studies revealed new AIS proteins and were a first start at defining the AIS proteome.
Response1: We appreciate the reviewer’s thoughtful comment. In response, we have simplified the text by removing the sentence(Line 96).
Comment 2: The reference for Kcnab2 in the list is incorrect. The referenced paper does not deal with Kvb2, but rather Kv2.1.
Response 2: Thank you for pointing out this error. We have carefully reviewed the reference and confirmed that the originally cited paper was indeed related to Kv2.1, not Kvb2 (Kcnab2). In response, we have replaced it with a more appropriate reference that specifically discusses Kvb2(Zhang W et al., Sci Adv 2025).
Comment 3: I suggest including the TRAAK/TREK-1 two-pore potassium channel in Fig. 1, and include the reference from Escobedo et al. (2024), JCB.
Response 3: We thank the reviewer for this valuable suggestion. In response, we have added the TRAAK (K2P4.1) two-pore potassium channel to Figure 2 and Table 2 to reflect its reported localization at the AIS. We have also included the appropriate reference to Escobedo et al. (2024, JCB) in both the main text and the reference list.
Reviewer 3 Report
Comments and Suggestions for Authors
The authors of this review used current axon initial segment (AIS) proteomic data and described in detail the interaction of ankyrin-G with some AIS proteins, in particular sodium and potassium ion channels/transporters, spectrins, actin fibers and microtubules. This is a well written overview that seeks to understand the proteome in the AIS and which may provide insight into the new regulatory mechanisms for controlling the structure and function of the AIS.
I have only few minor comments.
Minor Comments:
The genes of some of the proteins in Figure 2 are not present in Table 1, which is not explained. Not even some abbreviations, such as CAMSAP2 and NF-186, shown in Figure 2, are not explained. Also confusing is the fact that Table 1 contains a large number of AIS genes that are not mentioned in the article, and conversely some genes of AIS-specific proteins that are mentioned in the text, are not mentioned in the Table 1, e e.g. genes for Caspr2. Tag1, Nav1.6, Kv1 channels
Figure 2: βIV spectrins are in grey, not pink. Pink are βII spectrins, but these bind to ankyrin B.
Table 1: The authors should consider to italicize gene symbols in this table and the whole paper. For example, for "SCNA8A" in the text to Table 1, the cursive is required.
Author Response
Comments 1: The genes of some of the proteins in Figure 2 are not present in Table 1, which is not explained.
Response 1: Thank you for this important comment. We would like to clarify that not all AIS-related proteins shown in Figure 2 are listed in Table 1 because some of them did not rank highly in the proteomic analysis. This point has been explicitly described in the revised manuscript(Line 100).
Comments 2: Not even some abbreviations, such as CAMSAP2 and NF-186, shown in Figure 2, are not explained.
Response 2: We appreciate the reviewer’s careful observation. In response, we have added definitions for all abbreviations used in Figure 2, including CAMSAP2 and NF-186, in the bottom of figure legend to enhance clarity for readers(Line70). Additionally, we have included a comprehensive list of abbreviations with their definitions at the end of the article.
Comments 3: Also confusing is the fact that Table 1 contains a large number of AIS genes that are not mentioned in the article, and conversely some genes of AIS-specific proteins that are mentioned in the text, are not mentioned in the Table 1, e e.g. genes for Caspr2. Tag1, Nav1.6, Kv1 channels
Response 3: Thank you for this important comment. We would like to clarify that not all AIS-related proteins shown in Figure 2 are listed in Table 1 because some of them did not rank highly in the proteomic analysis. This point has been explicitly described in the revised manuscript(Line 100)
Comment 4: Figure 2: βIV spectrins are in grey, not pink. Pink are βII spectrins, but these bind to ankyrin B.
Response 4: We thank the reviewer for pointing out this error. We have corrected the figure to represent βIV spectrins in orange and βII spectrins in pink and αII spectrins in grey, in accordance with their proper molecular identity and binding partners. The figure legend has also been updated to clearly indicate this distinction.
Comment 5: Table 1: The authors should consider to italicize gene symbols in this table and the whole paper. For example, for "SCNA8A" in the text to Table 1, the cursive is required.
Response 5: We thank the reviewer for this important suggestion. In response, we have revised Table 1 and the main text to italicize all gene symbols, including SCN8A, in accordance with standard gene nomenclature guidelines. We appreciate the reviewer’s attention to detail and believe this correction improves the accuracy and professionalism of the manuscript.